# Hypoglycemic Effect of Polysaccharides from *Physalis alkekengi* L. in Type 2 Diabetes Mellitus Mice

**DOI:** 10.3390/biology13070496

**Published:** 2024-07-04

**Authors:** Yun Zhang, Minghao Wang, Peng Li, Ge Lv, Jing Yao, Lin Zhao

**Affiliations:** 1College of Food Engineering, Heilongjiang East University, Harbin 150066, China; mhwangbio@163.com (M.W.); 18845796004@163.com (P.L.); gracelvge@163.com (G.L.); yaojingnd@126.com (J.Y.); 2Quality & Safety Institute of Agricultural Products, Heilongjiang Academy of Agricultural Sciences, Harbin 150086, China; zhaolin78@126.com

**Keywords:** type 2 diabetes mouse model, *Physalis alkekengi* L. polysaccharides, hypoglycemic

## Abstract

**Simple Summary:**

Type 2 diabetes mellitus is harmful to patients. At present, the main treatment for type 2 diabetes mellitus is to control blood glucose by taking chemical synthesized drugs. However, taking drugs chronically will increase the risk of side effects. Therefore, this study uses polysaccharides extracted from natural plant *Physalis alkekengi* L. to conduct animal experiments and research whether it can alleviate the related symptoms of type 2 diabetes mellitus in mice and improve the health level of mice, so as to obtain a natural active ingredient that can assist or, to some extent, replace drugs to treat type 2 diabetes mellitus.

**Abstract:**

Type 2 diabetes mellitus (T2DM) is a common metabolic disease that adversely impacts patient health. In this study, a T2DM model was established in ICR mice through the administration of a high-sugar and high-fat diet combined with the intraperitoneal injection of streptozotocin to explore the hypoglycemic effect of polysaccharides from *Physalis alkekengi* L. After six weeks of treatment, the mice in the high-dosage group (800 mg/kg bw) displayed significant improvements in terms of fasting blood glucose concentration, glucose tolerance, serum insulin level, insulin resistance, and weight loss (*p* < 0.05). The polysaccharides also significantly regulated blood lipid levels by reducing the serum contents of total triglycerides, total cholesterol, and low-density lipoproteins and increasing the serum content of high-density lipoproteins (*p* < 0.05). Furthermore, they significantly enhanced the hepatic and pancreatic antioxidant capacities, as determined by measuring the catalase and superoxide dismutase activities and the total antioxidant capacity (*p* < 0.05). The results of immunohistochemistry showed that the *P. alkekengi* polysaccharides can increase the expression of GPR43 in mice colon epithelial cells, thereby promoting the secretion of glucagon-like peptide-1. In summary, *P. alkekengi* polysaccharides can help to regulate blood glucose levels in T2DM mice and alleviate the decline in the antioxidant capacities of the liver and pancreas, thus protecting these organs from damage.

## 1. Introduction

*Physalis alkekengi* L. (Chinese lantern) is an important perennial plant belonging to the Solanaceae family. It was first recorded in Shennong’s Materia Medica Classic and was also included in the Compendium of Materia Medica and the 2015 edition of the Pharmacopoeia of the People’s Republic of China. Its calyx and fruit have traditionally been consumed as Chinese herbal medicine and edible berries, with functions such as clearing heat, diuresis, and resolving phlegm [1,2]. Polysaccharides are one of the important active substances in *P. alkekengi* and exhibit various biological activities [3] while possessing minimal toxicity. They are widely present in animals and plants [4,5] and exert regulatory effects on blood glucose, lipid metabolism [6,7,8], intestinal flora structure [9,10], and antioxidant capacity [11,12,13]. Type 2 diabetes mellitus (T2DM), also known as non-insulin-dependent diabetes in the early stage, is a chronic metabolic disease caused by metabolic abnormalities. Because T2DM is commonly seen in adults aged 35–40 years old, it was formerly known as adult-onset diabetes, and it accounts for more than 90% of the total number of diabetes cases. The main characteristic features of T2DM include hyperglycemia, insulin resistance [14], and insulin deficiency. Common symptoms experienced by T2DM patients include thirst, hunger, frequent urination, and weight loss [15]. In addition to the direct effects on patient health, T2DM patients are also prone to a variety of complications [16], such as atherosclerosis [17], diabetic encephalopathy [18], cardiac insufficiency caused by diabetic cardiomyopathy [19], diabetic nephropathy [20], and diabetic retinopathy [21,22].

There is currently no effective cure for diabetes, and the harm associated with this condition can only be alleviated by reducing postprandial blood sugar. Common clinical methods for achieving this include insulin injections and oral hypoglycemic drugs. Although numerous hypoglycemic drugs have been developed, all have certain side effects [23]. Consequently, many researchers have turned their attention to natural substances, and the identification of natural active ingredients that can lower blood sugar with minimal side effects has emerged as a hot research topic. In particular, various plant-derived polysaccharides have been found to display significant hypoglycemic effects, such as those extracted from pumpkins by Wang et al. [24] and those extracted from *Codonopsis pilosula* by Liu et al. [25]. The administration of these two polysaccharide extracts to T2DM mice was found to reduce blood sugar and enhance glucose tolerance, regulate the blood lipid content, and improve the antioxidant levels in the liver and pancreas. However, only limited research has been conducted on the polysaccharides from *P. alkekengi*.

Preliminary in vitro experiments conducted by our research group indicated that polysaccharides extracted from *P. alkekengi* significantly inhibit the activity of α-glucosidase [26], thus having the potential to suppress carbohydrate metabolism in the human body and reduce blood glucose levels [27]. However, it has not yet been confirmed whether *P. alkekengi* polysaccharides exert the same hypoglycemic effect in vivo. In this work, a crude *P. alkekengi* polysaccharide extract was administered to T2DM mice via gavage. The model mice of type 2 diabetes mellitus were induced via a high-fat diet and streptozotocin. Because streptozotocin has a serious destructive effect on pancreatic β cells, this model can simulate the T2DM patients with serious conditions and even T2DM patients with insulin dependence. Such severe type 2 diabetes may increase the risk of heart disease and vascular disease and even develop into a life-threatening hyperosmolar hyperglycemic state (HHS). The purpose of this study is to research the alleviating effect of polysaccharides on T2DM, control the blood glucose levels of patients, and then reduce the risk of serious complications caused by type 2 diabetes. Therefore, this study will provide a possible treatment method other than chemical synthesized drugs for T2DM patients with more serious conditions and provide data and theoretical support for the future research and development of related health food.

## 2. Materials and Methods

### 2.1. Materials and Chemicals

#### 2.1.1. Preparation of Streptozotocin Solution

Sodium citrate buffer: Citric acid (2.1 g) was dissolved in distilled water (100 mL) to obtain solution A. Sodium citrate (2.94 g) was dissolved in distilled water (100 mL) to obtain solution B. Streptozotocin preparation solution: solution A and solution B were mixed at a ratio of 1:1.32, the pH value was adjusted to 4.2–4.5, and streptozotocin was prepared as needed.

#### 2.1.2. Polysaccharide Extraction

The crude polysaccharides were extracted from *P. alkekengi* according to the optimized method described in our previous work [26]. In brief, water extraction was performed using a material/liquid ratio of 1:24 g/mL, an extraction temperature of 93 °C, and an extraction time of 2.3 h.

### 2.2. Animal Experiments

#### 2.2.1. Animal Procurement and Ethics Statement

Eighty SPF-grade four-week-old male ICR mice, each weighing 25 ± 2 g, were purchased from Liaoning Changsheng Biotechnology Co., Ltd. (Liaoning, China), with a production license number of SCXK (Liao) 2020-001. All animal experiments were reviewed and approved by the Laboratory Animal Ethics Committee of Heilongjiang Oriental University under the approved protocol number Specific Pathogen-Free Rodent Management (KY2019-03).

#### 2.2.2. Establishment of the Mouse Model

After one week of acclimation under a normal diet (35% sucrose, 10% fat, casein 200 mg, cystine 3 mg, cornstarch 315 mg, maltodextrin 35 mg, cellulose 50 mg, Soybean oil 25 mg, lard oil 20 mg, mineral substance 10 mg, calcium hydrogen phosphate 13 mg, calcium carbonate 5.5 mg, potassium citrate 16.5 mg, cellulose mixture 10 mg, choline bitartrate 2 mg, food colorants 0.05 mg per 1000 mg), the majority of the mice (*n* = 70) were transitioned to a high-fat diet (20% sucrose, 45% fat, casein 233.06 mg, cystine 3.5 mg, cornstarch 84.83 mg, maltodextrin 116.53 mg, cellulose 58.26 mg, Soybean oil 29.13 mg, lard oil 206.84 mg, mineral substance 11.65 mg, calcium hydrogen phosphate 15.15 mg, calcium carbonate 6.41 mg, potassium citrate 19.23 mg, cellulose mixture 11.56 mg, choline bitartrate 2.33 mg, food colorants 0.058 mg per 1000 mg) to induce T2DM. In addition, a control group of randomly selected mice (*n* = 10) continued on the normal diet. Four weeks later, the mice on the high-fat diet were intraperitoneally injected with streptozotocin (100 mg/kg bw), and the mice on the normal diet were intraperitoneally injected with an equal amount of sodium citrate buffer. The mice were then monitored for 1–3 d for any signs of dull fur or listlessness. After the emergence of these symptoms, all of the mice were fasted for 12 h, and their tail blood was collected for blood glucose measurements. During the experimental period, the body weight, food intake, and water intake of the mice were tested once a week. The successful establishment of the model was confirmed by the changes in the body weight, food intake, and water intake of the mice, as well as the fasting blood glucose measurements. The observation of polyuria, excessive drinking, overeating, and weight loss and fasting blood glucose values exceeding 11.1 mmol/L were considered indicative of the successful induction of T2DM.

#### 2.2.3. Grouping and Feeding

T2DM was successfully established in 66 of the 70 mice on the high-fat diet, and one mouse died. The T2DM mice were randomly divided into five groups, with ten mice in each group. Five to seven mice per cage were injected intraperitoneally with sodium citrate buffer as the normal control group. The six groups were as follows: (i) low-dosage polysaccharide (LF) group, where T2DM mice were fed a high-fat diet and given 200 mg/kg bw of *P. alkekengi* polysaccharide extract via gavage; (ii) medium-dosage polysaccharide (MF) group, where T2DM mice were fed a high-fat diet and given 400 mg/kg bw of *P. alkekengi* polysaccharide extract via gavage; (iii) high-dosage polysaccharide (HF) group, where T2DM mice were fed a high-fat diet and given 800 mg/kg bw of *P. alkekengi* polysaccharide extract via gavage; (iv) positive control (PC) group, where T2DM mice were fed a high-fat diet and given 100 mg/kg bw metformin solution via gavage; (v) model control (NC) group, where T2DM mice were fed a high-fat diet and given an equal volume of physiological saline via gavage; and (vi) normal control (DC) group, where healthy mice were fed a normal diet and given an equal volume of physiological saline via gavage. Intragastric gavage was performed every day for six weeks starting from 1:00 p.m. each day, and the gavage volume was 0.2 mL. The laboratory was maintained at a constant temperature of 25 ± 1 °C with a relative humidity of 50–60% and a 12 h light/12 h dark cycle.

#### 2.2.4. Monitoring

The mental behavior of the mice and the color and glossiness of their fur were recorded every week. The mice were also weighed once per week to assess the changes in body weight.

#### 2.2.5. Sampling of Serum and Organs

At the end of the experiment, serum samples were obtained by anesthetizing the mice with chloral hydrate and drawing blood from the posterior carotid artery. The collected blood was allowed to stand for 30 min, then centrifuged at 3000 r/min for 15 min. The upper layer of serum was collected, divided into 2 mL cryotubes, and labeled. The serum samples were frozen at −80 °C.

After the blood collection, the mice were euthanized via cervical dislocation and dissected. The entire dissection process was performed under sterile conditions. The liver, pancreas, kidneys, thymus, and spleen were extracted; the attached fat was removed; and the organs were cleaned with physiological saline. After absorbing the excess saline using filter paper, the organs were weighed and stored at −80 °C [28].

### 2.3. Measurement of Fasting Blood Glucose and Glucose Tolerance

Measurements were performed according to the methods described by Wang et al. [29]. Every week, the mice were fasted for 12 h, then blood was collected from the tail vein and the fasting blood glucose levels were measured using a blood glucose meter (Roche, Ludwigsburg, Germany). The glucose tolerance of the mice was also determined 2 d prior to the end of the experiment. The fasting blood glucose values were first measured as above. All of the mice were then intraperitoneally injected with 10% glucose solution (2 g/kg bw), and the resulting blood glucose values were measured after 0.5, 1, and 2 h. These were used to calculate the area under the blood glucose time curve (AUC) values.

### 2.4. Measurement of Serum Biochemical Indicators

The serum concentrations of high-density lipoprotein (HDL-C), low-density lipoprotein (LDL-C), total cholesterol (TC), and total triglycerides (TG) were determined according to the instructions for the corresponding assay kits (Nanjing Jiancheng Bioengineering Institute, Nanjing, China).

The serum concentrations of glycosylated proteins and insulin were measured according to the instructions for the corresponding assay kits (Wuhan Enzyme-free Biotechnology Co., Ltd., Wuhan, China). The results were used to estimate insulin resistance using the homeostatic model assessment of insulin resistance (HOMA-IR).

### 2.5. Measurement of Organ Indices and Antioxidant Capacities

Since changes in organ weight may reflect related organ lesions, measuring organ weight can have attributes in assessing the health statuses of mice. So, in mice experiments, the organ index is usually measured to determine whether mice organs are normal or pathological. In this research, the organ indices were calculated for the liver, kidneys, and pancreas using the following formula: organ index (mg/g) = organ weight (mg)/body weight (g).

The livers and pancreases removed from the mice were homogenized using a high-speed frozen tissue homogenizer (Shanghai Sheyan Instrument Co., Ltd., Shanghai, China), and the antioxidant levels were measured using superoxide dismutase (SOD), catalase (CAT), and total antioxidant capacity (T-AOC) kits (Nanjing Jiancheng Bioengineering Institute, Nanjing, China).

### 2.6. Immunohistochemistry of Colon and Pancreatic Tissues

The colons and pancreases of mice were collected under sterile conditions and then fixed with 4% paraformaldehyde solution. After dehydration, we embedded them in paraffin and cut them into 5 μm coronal sections for immunohistochemical analysis. Colons coronary sections were incubated with primary antibodies against GLP-1 and GPR43 (ABclonal, Woburn, MA, USA), and pancreas coronary sections were incubated with primary antibodies against GLP-1 (ABclonal, USA) at 4 °C for 12 h. Then, the slices were incubated with a second antibody (1:400) (Thermo, Waltham, MA, USA) and visualized using diaminobenzidine (DAB) as the chromogen. We observed the sample under an optical microscope and took photos, then used Image-Pro Plus 6.0 to measure its optical density.

### 2.7. Statistical Analysis

The experimental data were analyzed using SPSS Statistics 17.0 software. *p* values of less than 0.05 were considered to indicate statistical significance. All bar charts in this work were generated using Origin 2019. Data are presented as the mean ± standard deviation.

## 3. Results

### 3.1. The Main Components and Molecular Structure of P. alkekengi Polysaccharides

In the preliminary research of the research group, cellulose DE-52 and Sephadex G-100 were used as purification materials to purify *P. alkekengi* polysaccharides using the chromatography column method. Three main components were obtained, including a deionized water elution component (Phy-1A), 0.1 mol/L NaCl elution component (Phy-2A), and 0.3 mol/L NaCl elution component (Phy-3A). The weight-averaged molecular weights of Phy-1A, Phy-1B and Phy-1C were determined to be 59 kDa, 9.8 kDa and 9.8 kDa via High-Performance Gel Permeation Chromatography (HPGPC).

The purified component Phy-1B obtained a single symmetrical peak with a peak area ratio of 100%. The results of nuclear magnetic resonance structural analysis showed that the main chain of Phy-1B was a glycosidic bond in the →2)-α-L-Rhaf-(1→4)-β-D-Galp-(1→4)-β-D-Galp-(1→[3)-β-D-Glcp-(1]2→3)-β-D-Glcp-(1→[4)-β-D-Glcp-(1]2→ structure, while the branches were α-L-Araf-(1→5)-α-L-Araf-(1→, β-D-Glcp-(1→4)-β-D-Xylp-(1→3)-β-D-Galp-(1→, β-D-Glcp-(1→6)-β-D-Glcp-(1→. The three fragments were connected to the main chain via the O-4 bond of →2,4)-α-L Rhaf-(1→), the O-6 bond of →4,6)-β-D Galp-(1→), and the O-6 bond of →3,6)-β-D Glcp-(1→), respectively (the above research results have not been published).

### 3.2. P. alkekengi Polysaccharides Alleviate Weight Loss in Mice

After four weeks, the mice fed the high-fat diet displayed significantly decreased body weights compared with the mice fed the normal diet (*p* < 0.05) (The data of average food intake of mice are showed in Table 1). In combination with the fasting blood glucose measurements, this indicated the successful induction of T2DM. Figure 1 shows the body weight changes for each group of mice over six weeks of administering the *P. alkekengi* polysaccharide extract at various dosages. After four weeks of the high-fat diet and streptozotocin induction, the T2DM mice exhibited significant weight loss compared with the DC group (*p* < 0.05). After six weeks of treatment, the DC group displayed the highest weight gain (14.54%) among the six groups. The weight gains for the PC, LF, MF, and HF groups were 5.01%, 1.46%, 1.83%, and 6.84%, respectively. Compared with the NC group, after six weeks of treatment, the weights of the T2DM mice in the PC and HF groups had significantly increased (*p* < 0.05), although they were still significantly lower than the weight observed for the DC group (*p* < 0.05). The weight changes for the mice in the MF and LF groups were not significant compared with the NC group (*p* > 0.05). These results indicate that a high dosage of *P. alkekengi* polysaccharides partially alleviated the weight loss typically observed for T2DM mice, but the mice still did not exhibit the normal weight gain expected for healthy mice.

### 3.3. P. alkekengi Polysaccharides Reduce Fasting Blood Glucose in Mice

Figure 2 presents the fasting blood glucose results obtained for the six groups. At week 0, the fasting blood glucose values for the NC, PC, LF, MF, and HF groups were 15.75 ± 2.79, 16.44 ± 3.78, 16.03 ± 3.32, 16.42 ± 2.04, and 16.26 ± 2.21 mmol/L, respectively. After six weeks of treatment, the corresponding values were 15.41 ± 1.11, 10.24 ± 1.32, 13.80 ± 2.64, 14.81 ± 1.85, and 11.13 ± 3.19 mmol/L, respectively. The initial fasting blood glucose values for the five groups of T2DM mice were similar without significant differences, and they were significantly higher than those recorded for the DC group (*p* < 0.01). Over the course of the experiment, the fasting blood glucose values remained essentially unchanged for the NC group, whereas those for the polysaccharide treatment groups exhibited an overall downward trend.

Compared with the NC group, the fasting blood glucose values for the HF group after six weeks of treatment with a high dosage of *P. alkekengi* polysaccharides were significantly reduced (*p* < 0.05), similar to the results observed for the PC group, although they remained higher than 11.1 mmol/L and were significantly different from those for the healthy mice in the DC group (*p* < 0.01). Although no significant reduction in fasting blood glucose values was observed for the LF group compared with the NC group (*p* > 0.05), the decrease with respect to week 0 within the LF group was found to be significant (*p* < 0.05).

### 3.4. P. alkekengi Polysaccharides Regulate Glucose Tolerance in Mice

The glucose tolerance of each group of mice was assessed by measuring the changes in blood glucose concentration following the intraperitoneal injection of glucose solution (Figure 3a) and calculating the AUC values based on the resulting data (Figure 3b).

Previous studies have demonstrated that there is a close relationship between fasting blood glucose levels and glucose tolerance [30]. In Figure 3a,b, it can be seen that the NC group displayed significant increases in both the blood glucose concentrations at every time point and the AUC values compared with the DC group following the injection of glucose (*p* < 0.05), indicating abnormal glucose tolerance for the T2DM mice. Compared with the NC group, after six weeks of treatment, the blood glucose and AUC values of the T2DM mice in the 2 h after glucose injection were significantly reduced for the HF and PC groups (*p* < 0.05). Although these parameters did not reach the low levels observed for the DC group compared with the MF and LF groups (no significant difference was observed between the MF, LF, and NC groups), the HF group still displayed some influence of the treatment in terms of regulating the blood glucose and glucose tolerance of the mice, indicating that a high dosage of *P. alkekengi* polysaccharides can regulate the hyperglycemia and glucose tolerance abnormalities of T2DM mice.

### 3.5. P. alkekengi Polysaccharides Reduce Insulin Resistance

Table 2 shows the HOMA-IR values calculated for each group of mice based on the fasting blood glucose and serum insulin concentrations. It can be seen that the serum insulin concentrations and HOMA-IR values were significantly elevated for the NC group compared with the DC group (*p* < 0.05), indicating that the T2DM mice were in a state of insulin resistance and could not effectively use insulin to regulate blood glucose. Compared with the NC group, the serum insulin concentrations were significantly reduced for the MF and HF groups (*p* < 0.05), the HOMA-IR values were significantly reduced for the HF group (*p* < 0.05), and the HOMA-IR values were slightly reduced for the LF and MF groups, but these changes were not significant (*p* > 0.05). The serum insulin concentrations of the T2DM mice in the HF group after six weeks of treatment were similar to those observed for the PC group (*p* > 0.05), and there was no significant difference (*p* > 0.05) compared with the healthy mice in the DC group. However, there was still a significant difference in the HOMA-IR values compared with the healthy mice in the DC group. These results indicate that a high dosage of *P. alkekengi* polysaccharides can help T2DM mice to effectively utilize insulin and reduce their insulin resistance, thus displaying similar effects to metformin, although the HOMA-IR values did not reach those observed for healthy mice.

### 3.6. P. alkekengi Polysaccharides Regulate Blood Lipid Levels in Mice

Diabetic patients generally exhibit some symptoms of lipid metabolism disorder, and clinical research has demonstrated that TG, TC, HDL-C, and LDL-C values are closely related to cardiovascular diseases and atherosclerosis, where long-term lipid metabolism disorder increases the probability of cardiovascular diseases in diabetic patients [31,32]. After six weeks of treatment, the aforementioned blood lipid indicators were measured, and the results are listed in Table 3. It can be seen that the T2DM mice in the NC group displayed significantly elevated TC and TG values compared with the DC group (*p* < 0.05), indicating that the T2DM mice exhibited symptoms of lipid metabolism disorder and increased blood lipid levels. Compared with the NC group, after six weeks of gavage with *P. alkekengi* polysaccharides, the TC and TG values were significantly reduced for the HF, MF, and LF groups (*p* < 0.05). Similar results were observed for the PC group, and there was no significant difference compared with the DC group (*p* > 0.05). These findings indicate that *P. alkekengi* polysaccharides exert a beneficial effect on regulating blood lipid levels in T2DM mice, and this effect was dose-dependent.

### 3.7. P. alkekengi Polysaccharides Improve Hepatic Antioxidant Capacity in Mice

The liver is an important organ for regulating blood glucose levels, and the CAT, SOD, and T-AOC values are useful indicators for evaluating the antioxidant capacity of the body. These parameters were measured separately for the livers removed from each group of mice, and the results are presented in Figure 4. Compared with the DC group, the hepatic SOD, CAT, and T-AOC values were significantly reduced for the NC group (*p* < 0.05), indicating a decrease in the antioxidant capacity of the liver in the T2DM mice compared with the healthy mice.

In Figure 4a, it can be seen that the CAT activities in the livers of the T2DM mice in the LF and HF groups were significantly increased compared with the value of 81.41 ± 9.95 mmol/gprot observed for the NC group (*p* < 0.01), while the CAT activity increased less for the MF group. Compared with the PC group, the hepatic CAT activities of the LF and HF groups were higher, although the difference was not significant (*p* > 0.05).

In Figure 4b, it can be seen that the SOD activities in the livers of the T2DM mice in all three groups treated with *P. alkekengi* polysaccharides were significantly increased compared with the value of 17.85 ± 2.34 mmol/gprot observed for the NC group (*p* < 0.05). In particular, the hepatic SOD activity displayed a very significant increase for the HF group, with a difference of 20.59 ± 1.62 mmol/gprot (*p* < 0.01).

In Figure 4c, it can be seen that the LF, MF, HF, and PC groups displayed significantly increased T-AOC values in the livers compared with the value of 0.05 ± 0.009 mmol/gprot recorded for the NC group (*p* < 0.05). In particular, the hepatic T-AOC values exhibited very significant increases for the LF and PC groups (*p* < 0.01).

These results indicate that *P. alkekengi* polysaccharides can enhance the CAT and SOD activities and improve the T-AOC values of the livers of T2DM mice, thereby improving the antioxidant capacity of the liver and the metabolic capacity of the body. The observed effects were similar to those of metformin.

### 3.8. P. alkekengi Polysaccharides Improve Pancreatic Antioxidant Capacity in Mice

The CAT, SOD, and T-AOC values were also determined for the pancreases removed from each group of mice. Oxidative stress plays an important role in diabetes and its complications, and the pancreas is especially vulnerable through damage to the pancreatic β cells. Therefore, the pancreatic antioxidant capacity plays a crucial role.

As shown in Figure 5, there was no significant difference in the pancreatic SOD activities or T-AOC values between the NC and DC groups (*p* > 0.05), whereas the CAT activity was very significantly reduced in the NC group compared with the DC group (*p* < 0.01).

The pancreatic SOD activities were significantly increased for the MF and HF groups compared with the value of 124.31 ± 11.24 mmol/gprot recorded for the NC group (*p* < 0.05). In particular, the SOD activity was very significantly increased for the MF group compared with the NC group (*p* < 0.01), and the activity for the MF group was not significantly different from that for the PC group (*p* > 0.05). This indicates that the medium dosage of *P. alkekengi* polysaccharides increased the pancreatic SOD activity in the T2DM mice, which was similar to the therapeutic effect of metformin.

The pancreatic T-AOC values were significantly increased for all four treatment groups (PC, LF, MF, and HF) compared with the value of 0.059 ± 0.008 mmol/gprot observed for the NC group (*p* < 0.05), with the largest increase recorded for the HF group (*p* < 0.01), which exhibited a T-AOC value of 0.085 ± 0.0283 mmol/gprot.

The pancreatic CAT activity was significantly increased for the MF group compared with the value of 0.74 ± 0.13 mmol/gprot obtained for the NC group (*p* < 0.01). The value for the former group reached 1.1270 ± 0.20 mmol/gprot, which was not significantly different from that recorded for the PC group. This indicates that *P. alkekengi* polysaccharide extract alleviated the decrease in pancreatic CAT activity in the T2DM mice, and the observed effect was similar to that for metformin.

After six weeks of high-dosage treatment with *P. alkekengi* polysaccharides, the pancreatic T-AOC value and SOD activity were significantly improved to a similar extent as those observed for the PC group. These findings show that *P. alkekengi* polysaccharides can improve pancreatic antioxidant capacity in T2DM mice and alleviate the damage caused to pancreatic β cells.

### 3.9. P. alkekengi Polysaccharides Regulating the Expression Levels of T2DM Related Proteins in Mice

Immunohistochemical methods were used to detect the expression levels of GLP-1 and GPR43 in mice colon tissue.

GPR43 is a short-chain fatty acid (SCFA) receptor that can receive stimulation from SCFAs. When activated, GPR43 located in intestinal epithelial cells can promote its secretion of glucagon like peptide-1 (GLP-1) by regulating downstream-related pathways.

As shown in Figure 6, the secretion of GLP-1 in the colon tissue of mice in the HF group increased compared to that of the NC group, and the effect was similar to that of health mice in DC group.

As can be seen in Figure 7, the expression level of GPR43 in the colon tissue of mice in the HF group was higher than that of mice in the NC group.

The quantity analysis data of immunohistochemistry in Figure 8 shows that the integrated optical density of GLP-1 and GPR43 in HF group is significantly increased compare to that in NC group (*p* < 0.05).

These results suggest that the content of SCFAs in the intestine of mice may have increased, thereby increasing the expression of SCFA receptors, thus achieving the result of promoting the secretion of GLP-1 by the intestinal epithelial cells of mice.

### 3.10. P. alkekengi Polysaccharides Increas GLP-1 Content in Mouse Pancreatic Tissue

The immunohistochemical results in mice pancreatic tissue (Figure 9) showed that the GLP-1 content in the pancreatic tissue of mice treated with *P. alkekengi* polysaccharides was increased compared to the NC group, from the quantitative analysis data in Figure 10, it can be seen that this effect is extremely significant (*p* < 0.01), indicating that GLP-1 secreted by mice colon tissue can be successfully transferred to pancreatic tissue in some way, which may further affect mice pancreatic tissue. In addition, this result is consistent with the secretion levels of GLP-1 in each group of mice colon tissue, as the HF group has the most significant effect, making GLP-1 levels close to healthy mice in the DC group.

### 3.11. P. alkekengi Polysaccharides Decrease the Liver and Kidney Indices in T2DM Mice

The organ indices for the liver, kidneys, and pancreas with respect to the body weight were calculated for each group. The results are presented in Table 4. After treatment with *P. alkekengi* polysaccharides, the liver index decreased with increasing polysaccharide dosage. No significant difference was observed between the LF group and the DC group (*p* > 0.05), indicating that the low dosage of the *P. alkekengi* polysaccharide extract afforded a similar liver index to that observed for normal mice. The kidney index was significantly higher for the NC group than for the DC group (*p* < 0.05), and it also decreased with increasing polysaccharide dosage. The MF and HF groups displayed significant decreases compared with the NC group (*p* < 0.05), and there was no significant difference compared with the PC and DC groups (*p* > 0.05).

## 4. Discussion

Despite the harmful effects of T2DM on patient health, there is currently no effective cure for this disease. Controlling blood glucose using drugs is one of the more effective methods, but these drugs are typically associated with side effects [23,33]. In recent years, numerous studies have confirmed that polysaccharides extracted from natural foods can help to regulate blood glucose levels in T2DM mice and reduce the risk of organ damage and other complications caused by T2DM [34,35,36,37]. This suggests that polysaccharides have the potential to treat T2DM and may serve as non-pharmacological interventions for controlling blood glucose levels in T2DM patients.

In this work, the high-dosage administration of *P. alkekengi* polysaccharides exhibited a significant hypoglycemic effect in T2DM mice, significantly reducing the fasting blood glucose concentration and improving glucose tolerance (*p* < 0.05), and this effect was similar to that observed for the PC group using metformin (*p* > 0.05). Chen et al. demonstrated that the polysaccharides extracted from *Physalis pubescens* displayed a prospective hypoglycemic effect in streptozotocin-induced T2DM mice [38]. In addition, Yu et al. reported that the polysaccharides extracted from the persistent calyx of *P. alkekengi* exhibited a hypoglycemic effect in T2DM mice [39], in accordance with the results obtained in the current work. Previous research has shown that inhibiting the activity of α-glucosidase has a positive effect on blood glucose control [40]. Many studies have confirmed that α-glucosidase plays an important role as a therapeutic target in the treatment of T2DM [41,42]. The study by Chen et al. not only confirmed that polysaccharides can display the effect of regulating blood glucose levels in T2DM mice [38] but also demonstrated a strong inhibitory effect of the polysaccharide extracts on α-glucosidase activity in vitro. Previous in vitro results obtained by our research group also indicated that *P. alkekengi* polysaccharides can inhibit α-glucosidase activity [26]. In this study, the results demonstrated that *P. alkekengi* polysaccharides can significantly reduce fasting blood glucose and improve glucose tolerance in T2DM mice. Therefore, it appears that *P. alkekengi* polysaccharides help to regulate blood glucose levels in T2DM mice by suppressing the activity of α-glucosidase.

Diabetic patients generally display some symptoms of lipid metabolism disorder. Research has shown that TG, TC, HDL-C, and LDL-C values are all related to cardiovascular diseases and atherosclerosis, where long-term lipid metabolism disorder increases the risk of cardiovascular diseases in diabetic patients [32] and even leads to higher mortality for T2DM patients [43,44]. In addition, HDL-C can inhibit the oxidative modification of LDL-C and reduce thrombus formation, while a higher LDL-C content increases the incidence of atherosclerosis, and HDL-C can also reduce the TC level in the blood by promoting its metabolism in the vascular walls of atherosclerotic patients [34]. In this study, an appropriate dosage of *P. alkekengi* polysaccharides exhibited the ability to regulate blood lipids. Specifically, the high dosage administered to the HF group exerted marked regulatory effects on the TG, TC, LDL-C, and HDL-C levels in the T2DM mice, significantly reducing the serum contents of TG, TC, and LDL-C (*p <* 0.05) and significantly increasing the HDL-C content (*p* < 0.05). On the basis of these results, it can be inferred that a high dosage of *P. alkekengi* polysaccharides has a positive effect on regulating lipid metabolism by increasing the HDL-C content while reducing the TG, TC, and LDL-C contents, thereby having the potential to ameliorate lipid metabolism disorder in diabetic patients. Lipid metabolism disorder can lead to an increase in the concentration of free fatty acids, which have been found to induce the endoplasmic reticulum stress response of pancreatic β cells and promote the production of reactive oxygen species that can damage these cells [45]. Considering the regulatory effect that *P. alkekengi* polysaccharides exert on lipid metabolism in T2DM mice, it can be speculated that these polysaccharides may alleviate the increase in free fatty acid concentrations resulting from lipid metabolism disorder, thus reducing the endoplasmic reticulum stress response of pancreatic β cells and reducing the corresponding generation of reactive oxygen species, thus having the potential to protect pancreatic β cells and improve blood glucose regulation.

Previous research has also shown that oxidative stress is one of the important factors leading to the decline of pancreatic β cell function [46]. Oxidative stress involves the production of abnormally high amounts of reactive oxygen species and reactive nitrogen species, where the excess reactive oxygen species in particular are known to directly damage pancreatic β cells [47]. However, the antioxidant activities of CAT and SOD can reduce harmful oxidizing substances in organisms. In particular, CAT decreases the concentration of hydrogen peroxide, thereby suppressing the formation of free radicals and lipid peroxides. Meanwhile, SOD has the ability to convert superoxide anion radicals into hydrogen peroxide, which can then be further transformed into harmless water by CAT. In this study, the results obtained from assays of antioxidant enzyme activity and total antioxidant capacity revealed that the high-dosage administration of *P. alkekengi* polysaccharides to the HF group significantly increased the CAT and SOD activities and T-AOC values in both the livers and pancreases of T2DM mice (*p <* 0.05). This may reduce the risk of oxidative stress causing damage to pancreatic and hepatic cells. In accordance with these results, Zhou et al. fed black onion polysaccharide extracts to T2DM mice suffering from alloxan-induced liver and kidney damage and found that the polysaccharides effectively increased the SOD activity, reduced the reactive oxygen species and dialdehyde contents, and alleviated oxidative stress, thus protecting the liver and kidney cells of the mice [48]. This demonstrates that polysaccharides can protect the organs of T2DM mice by regulating antioxidant capacity. Therefore, it is speculated that the enhanced SOD activity observed in this study following the administration of *P. alkekengi* polysaccharides could reduce the content of reactive oxygen species that can damage pancreatic β cell function, ultimately protecting pancreatic cells and restoring their ability to regulate blood glucose in T2DM mice.

Insulin resistance refers to a decrease in the sensitivity and/or reactivity of insulin signal transduction, resulting in increased glucose absorption after the ingestion or administration of glucose [49]. The insulin resistance index HOMA-IR is an effective indicator of insulin resistance [50]. T2DM patients often exhibit higher HOMA-IR values, which means that they are in a state of insulin resistance. Numerous studies have demonstrated that insulin resistance is associated with a variety of diseases, including obesity, metabolic fatty liver disease, cardiovascular diseases, cancer, and so on [51]. The administration of a high dosage of *P. alkekengi* polysaccharides to the HF group significantly reduced the observed HOMA-IR values (*p* < 0.05). Although these did not reach the levels observed for the healthy mice (DC group), the effect was similar to that of metformin (*p >* 0.05). Thus, *P. alkekengi* polysaccharides displayed the ability to alleviate insulin resistance in the T2DM mice, which is consistent with the results previously observed for polysaccharides extracted from inulin, *Ganoderma lucidum*, and tea [52,53]. Consequently, *P. alkekengi* polysaccharide extract may reduce the risk of diseases related to insulin resistance and exert a positive effect on blood glucose regulation.

Recent studies have revealed that gut microbiota plays a crucial role in the treatment of T2DM [54]; meanwhile, researchers have pointed out that polysaccharides have the effect of regulating intestinal microbiota [55]. Corresponding to this fact, the relevant research results of our group showed that *P. alkekengi* polysaccharides have the function of regulating the composition of intestinal microbiota, such as increasing the abundance of *Akkermansia* muciniphila in the intestine, regulating the ratio of Firmicutes/Bacteroides, etc. (research results have not been published). The improvement of gut microbiota can stimulate the production of GLP-1 by intestinal epithelial cells, which, in turn, acts on the GLP-1 receptor (GLP-1r) of the pancreatic β cell and inhibits apoptosis in the pancreatic β cell. Furthermore, a series of positive effects will be activated, such as the activation of insulin synthesis-related genes or stimulation of insulin secretion [56]. The results of this study indicate that *P. alkekengi* polysaccharides have a protective effect on the pancreases of T2DM mice, indicating that *P. alkekengi* polysaccharides may exert this effect through the above mechanisms, thereby improving blood glucose metabolism in T2DM mice.

However, due to the limitations of the T2DM mouse model constructed with streptozotocin, it is not possible to fully reconstruct the human T2DM state, and the gavage dose of polysaccharides in mice is different from the intake dosage in humans. Can the converted human dose have the same therapeutic effect as it does in mice? With this question remaining unanswered, further exploration is needed in the future.

## 5. Conclusions

The results obtained in this study demonstrate that *P. alkekengi* polysaccharides have a significant effect on regulating blood sugar levels in T2DM mice, as well as alleviating the symptoms of weight loss, impaired glucose tolerance, and insulin resistance. These properties were similar to those observed for metformin. Therefore, *P. alkekengi* polysaccharides can exert hypoglycemic and related beneficial health effects in the body, providing a solid foundation for the future development of functional foods and health products.

## Figures and Tables

**Figure 1 biology-13-00496-f001:**
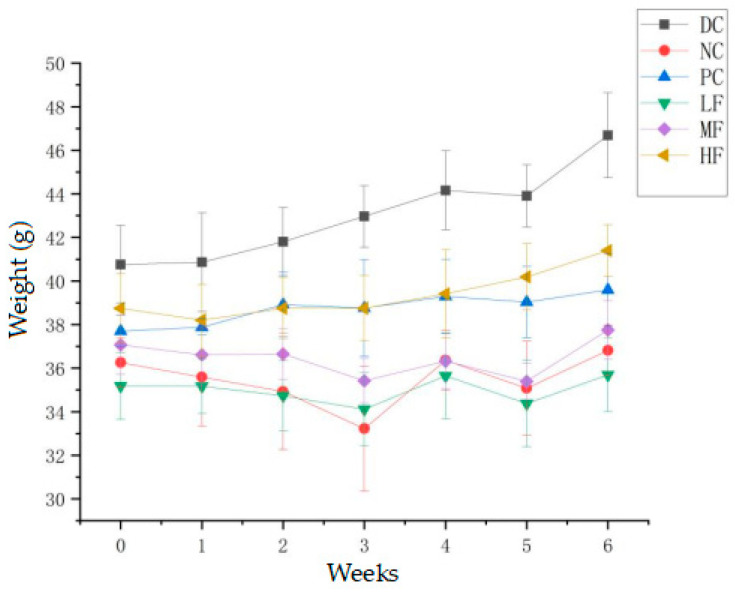
Effect of *P. alkekengi* polysaccharides on the body weights of T2DM mice. Data are presented as the mean ± SD, N = 10 in each group. Throughout the figures, DC denotes the normal control group, NC denotes the model control group, PC denotes the positive control group (metformin), LF denotes the low-dosage polysaccharide group (200 mg/kg bw), MF denotes the medium-dosage polysaccharide group (400 mg/kg bw), and HF denotes the high-dosage polysaccharide group (800 mg/kg bw).

**Figure 2 biology-13-00496-f002:**
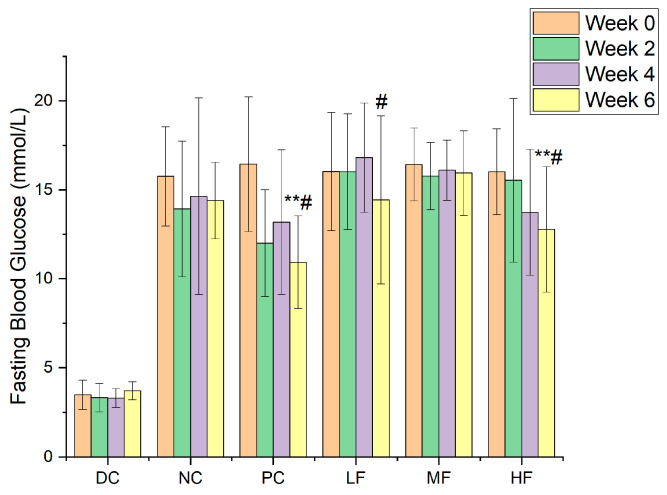
The variations in fasting blood glucose values for the six groups. Data are presented as the mean ± SD, N = 10 in each group. In the figure, ** indicates a significant difference with respect to the NC group (*p* < 0.01), while # indicates a significant difference within the group with respect to week 0 (*p* < 0.05).

**Figure 3 biology-13-00496-f003:**
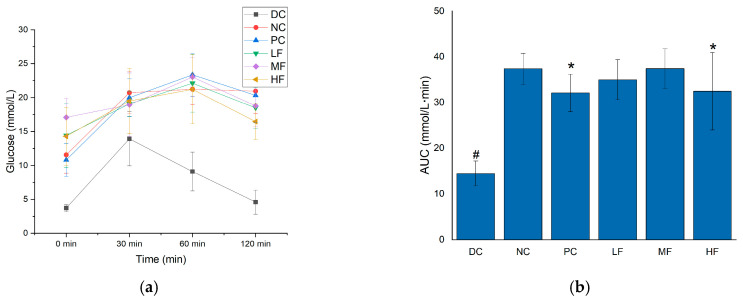
Effects of *P. alkekengi* polysaccharides on regulating glucose tolerance in T2DM mice: (**a**) blood glucose concentrations and (**b**) AUC values. Data are presented as the mean ± SD; N = 10 in each group. In the figure, # indicates a significant difference between the NC group and the DC group, while * indicates a significant difference with respect to the NC group (*p* < 0.05).

**Figure 4 biology-13-00496-f004:**
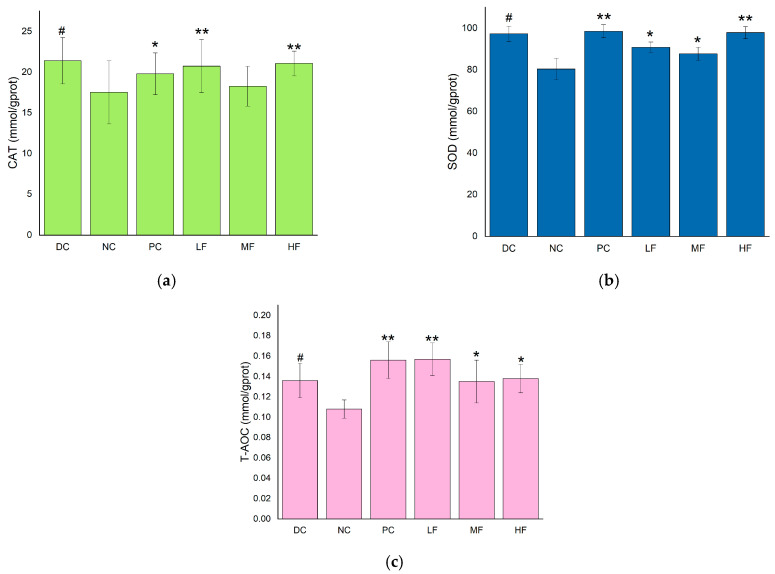
Effects of *P. alkekengi* polysaccharides on hepatic antioxidant activity in T2DM mice: (**a**) CAT activity, (**b**) SOD activity, and (**c**) T-AOC values. Data are presented as the mean ± SD; N = 5 in each group. In the figure, * indicates a significant difference with respect to the NC group (*p* < 0.05), ** indicates a very significant difference with respect to the NC group (*p* < 0.01), and # indicates a significant difference with respect to the DC group (*p* < 0.05).

**Figure 5 biology-13-00496-f005:**
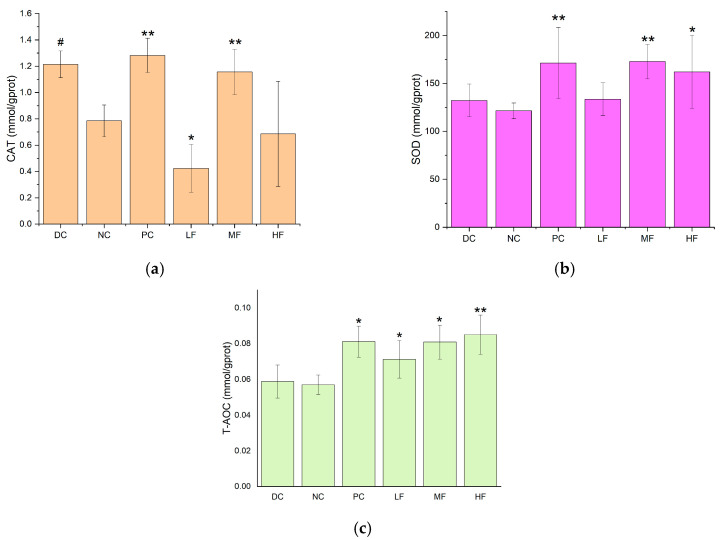
Effects of *P. alkekengi* polysaccharides on pancreatic antioxidant activity in T2DM mice: (**a**) CAT activity, (**b**) SOD activity, and (**c**) T-AOC values. Data are presented as the mean ± SD, N = 5 in each group. In the figure, * indicates a significant difference with respect to the NC group (*p* < 0.05), ** indicates a very significant difference with respect to the NC group (*p* < 0.01), and # indicates a significant difference with respect to the DC group (*p* < 0.05).

**Figure 6 biology-13-00496-f006:**
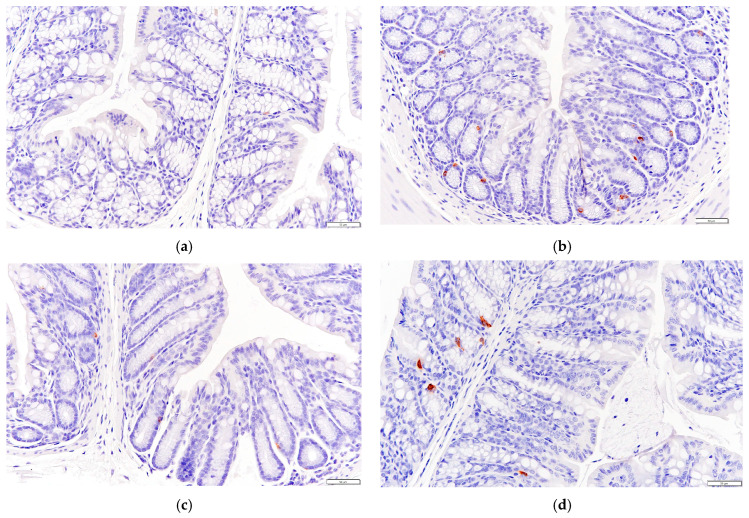
Effects of *P. alkekengi* polysaccharides on the expression level of GLP-1 in colon tissue in T2DM mice: (**a**) NC group, (**b**) DC group, (**c**) PC group, and (**d**) HF group. Display multiple is 100×.

**Figure 7 biology-13-00496-f007:**
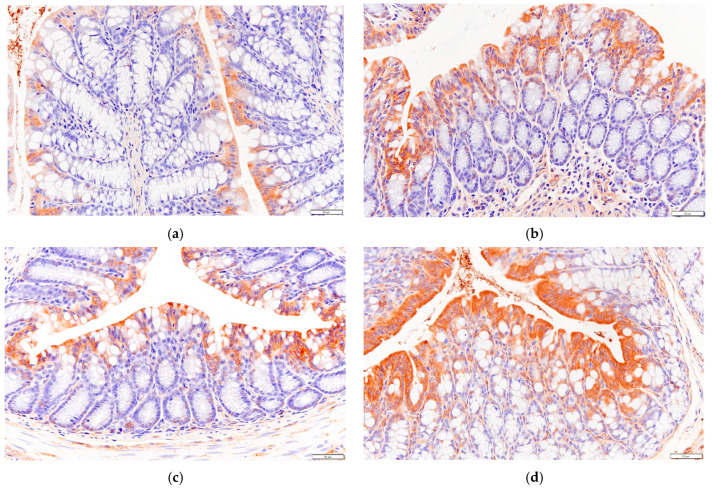
Effects of *P. alkekengi* polysaccharides on the expression level of GPR43 in colon tissue in T2DM mice: (**a**) NC group, (**b**) DC group, (**c**) PC group, and (**d**) HF group. Display multiple is 100×.

**Figure 8 biology-13-00496-f008:**
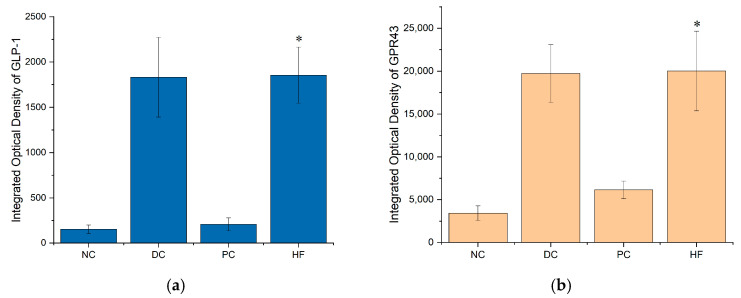
Integrated optical density (IOD) of GLP-1 and GPR43: (**a**) IOD of GLP-1; (**b**) IOD of GPR43. Data are presented as the mean ± SD; N = 5 in each group. * indicates *p* < 0.05 versus NC group.

**Figure 9 biology-13-00496-f009:**
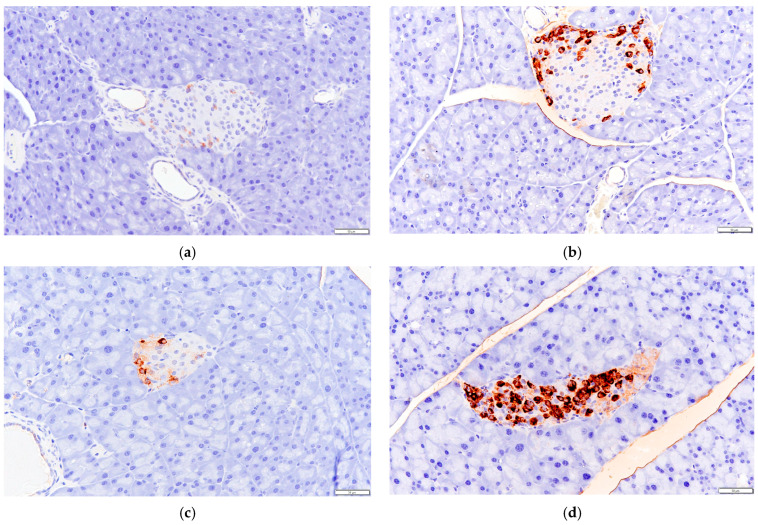
Effects of *P. alkekengi* polysaccharides on the content of GLP-1 in pancreatic tissue in T2DM mice: (**a**) NC group, (**b**) DC group, (**c**) PC group, and (**d**) HF group. Display multiple is 100×.

**Figure 10 biology-13-00496-f010:**
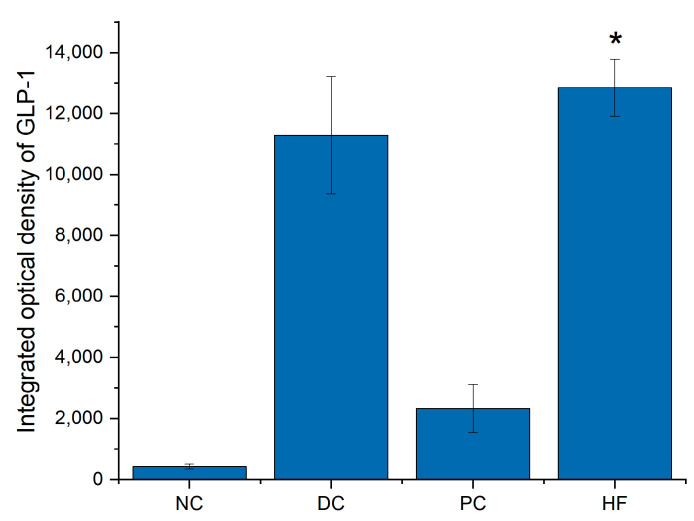
Integrated optical density (IOD) of GLP-1 in pancreatic tissue. Data are presented as the mean ± SD; N = 5 in each group. * indicates *p* < 0.01 versus NC group.

**Table 1 biology-13-00496-t001:** Average food intake of mice.

Group	Week					
	1	2	3	4	5	6
DC	4.86 ± 1.02	6.09 ± 1.24	5.18 ± 0.77	4.93 ± 0.55	5.75 ± 1.04	5.80 ± 1.59
NC	5.42 ± 0.33	6.02 ± 0.56	5.68 ± 1.20	5.45 ± 0.73	5.06 ± 0.84	5.47 ± 1.42
PC	5.70 ± 0.86	6.57 ± 1.12	6.30 ± 0.56	5.66 ± 1.09	5.71 ± 1.28	6.17 ± 1.32
LF	6.26 ± 1.11	6.63 ± 0.47	4.02 ± 1.02	5.25 ± 0.89	4.95 ± 0.98	4.76 ± 0.93
MF	5.83 ± 0.86	6.46 ± 0.92	5.09 ± 0.79	5.79 ± 1.23	6.31 ± 1.50	6.04 ± 1.24
HF	6.19 ± 1.03	7.25 ± 0.24	6.51 ± 1.21	7.07 ± 0.69	7.12 ± 1.03	7.08 ± 1.52

**Table 2 biology-13-00496-t002:** The effects of *P. alkekengi* polysaccharides on serum insulin concentrations and HOMA-IR values in T2DM mice ^1^.

Group	Insulin (mIU/L)	HOMA-IR
DC	40.86 ± 1.66 ^bc^	6.55 ± 0.65 ^c^
NC	42.98 ± 1.31 ^a^	29.39 ± 1.38 ^a^
PC	39.84 ± 1.12 ^c^	18.95 ± 1.91 ^b^
LF	43.72 ± 1.14 ^a^	29.02 ± 2.93 ^a^
MF	41.61 ± 1.39 ^b^	27.29 ± 2.53 ^a^
HF	40.10 ± 1.26 ^c^	22.03 ± 4.81 ^b^

^1^ Different superscript letters indicate significant differences between groups (*p* < 0.05). Mean values with the same subscript letter are not significantly different.

**Table 3 biology-13-00496-t003:** Effects of *P. alkekengi* polysaccharides on blood lipid levels in T2DM mice ^1^.

Group	TC (mmol/L)	TG (mmol/L)	LDL-C (mmol/L)	HDL-C (mmol/L)
DC	2.36 ± 0.28 ^b^	0.69 ± 0.11 ^b^	0.53 ± 0.06 ^ab^	0.67 ± 0.21 ^a^
NC	3.28 ± 0.70 ^a^	0.89 ± 0.13 ^a^	0.61 ± 0.13 ^a^	0.39 ± 0.10 ^c^
PC	2.40 ± 0.36 ^b^	0.73 ± 0.24 ^ab^	0.55 ± 0.05 ^ab^	0.62 ± 0.17 ^ab^
LF	2.65 ± 0.77 ^b^	0.68 ± 0.20 ^b^	0.58 ± 0.05 ^ab^	0.46 ± 0.20 ^abc^
MF	2.62 ± 0.60 ^b^	0.67 ± 0.21 ^b^	0.55 ± 0.07 ^ab^	0.40 ± 0.25 ^bc^
HF	2.34 ± 0.36 ^b^	0.58 ± 0.15 ^b^	0.51 ± 0.04 ^b^	0.59 ± 0.15 ^ab^

^1^ Different superscript letters indicate significant differences between groups (*p* < 0.05). Mean values with the same subscript letter are not significantly different.

**Table 4 biology-13-00496-t004:** Effects of *P. alkekengi* polysaccharides on the organ indices in T2DM mice ^1^.

Group	Liver Index	Kidney Index	Pancreas Index
DC	48.42 ± 2.28 ^b^	14.51 ± 0.88 ^b^	4.66 ± 0.70 ^b^
NC	45.37 ± 2.44 ^bc^	17.19 ± 1.06 ^a^	4.73 ± 0.93 ^ab^
PC	51.60 ± 5.10 ^a^	15.58 ± 1.03 ^b^	3.73 ± 0.23 ^c^
LF	47.61 ± 2.04 ^bc^	16.74 ± 1.08 ^a^	5.55 ± 1.11 ^a^
MF	44.89 ± 2.49 ^cd^	15.31 ± 1.24 ^b^	4.60 ± 0.80 ^b^
HF	42.07 ± 4.12 ^d^	12.52 ± 5.00 ^b^	3.99 ± 0.85 ^bc^

^1^ Different superscript letters indicate significant differences between groups (*p* < 0.05). Mean values with the same subscript letter are not significantly different.

## Data Availability

All data dealing with this study are reported in this paper.

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
