# Peer review of "Hypoglycemic Effect of Polysaccharides from Physalis alkekengi L. in Type 2 Diabetes Mellitus Mice"

_biology, 2024, doi:10.3390/biology13070496_

Round 1

Reviewer 1 Report (Previous Reviewer 2)

Comments and Suggestions for Authors

The authors argued that the therapeutic effects of Physalis alkekengi L. for type 2 DM was due to preventing oxidative stress and activation of GPR43 in colon. However, I have some questions about this stimulation.

1. While some significant differences were seen, I think the anti-oxidative stress effects of the Physalis alkekengi L was not so strong (Figure 4). There were few differences among the groups in serum insulin level (Table 3). In my understanding, the authors consider that Physalis alkekengi L could prevent damage in islets by preventing oxidative stress in islets. If so, the assessment of islets is necessary, at least histology. In my opinion, the therapeutic effects of Physalis alkekengi L is due to activation of GPR43 in colon, not by anti-oxidative effects.

As minor, please include quantification data of expression of GLP-1 and GPR43. What is the liver, kidney and pancreas "index"?

Author Response

Reviewer 2 Report (New Reviewer)

Comments and Suggestions for Authors

The manuscript by Zhang et al., was a pleasure to read. The authors claim the hypthesis that Physalis extract (specifically its polysaccharides) have a broad range of therapeutically valuable effects and aim to apply such to diabetes. Using a HFD+STZ mouse model, the authors investigate Low, Medium and High amounts of PAL extract and how this influence a broad range of metabolic endpoints.

The authors show that especially High doses of PAL extract are beneficial for diabetic weight loss, glucose tolerance and fasting blood glucose, insulin, pancreatic anti-oxidant responeses and GLP-1 expression in the gut lumen.

While the data are intriguing, the study lacks a couple of points that need to be addressed to improve the quality and merit of the study presented.

Major comments:

- The study employs a HFD + high dose STZ model. This model specifically aims to recapitualate insulin DEPENDANT type 2 diabetes. The introduction specifically fails to outline the population that this study aims to address.

The introduction also has to be overhauled in respect to diabetes and the different etiologies. Weight loss is NOT a major concern for Type 2 Diabetics but rather the exact opposite etc. Please revise this and clarfiy exactly what you are trying to study and why the model chosen is appropriate. Also clarify limitations for this model and thus the study.

- The graphs displayed are below publication quality - improve resolution please.

- Improve Figure legends as well. Name the N's per bar graph and statistics and if its SEM or SD or whatever was used. 

- Generally speaking the graphs dont seem like there would be significance givent he broad error bars. It would help to display individual data points or violin plots to emphasize data distribution in order to argue IN FAVOR of the data presented.

- The study is HIGHLY exploratory, which is great and has merit. But, the authors should voice that throughout the text. The PAL extract used is a mix of countless compounds that all do differnt things. You can not jsut argue that this will be a viable therapeutic without performing MULIT-OMIC studies to find active compounds repsonsible for i.e. anti ROS or insulin secretion. Revise accordingly please.

Minor comments:

- Minor editing of language required. Do not use non-scientific expressions such as GOOD/BAD. I.e. signficantly increased amount in A compared to B instead of better in A compared to B.

Comments on the Quality of English Language

Minor editing of language. Do not use non-scientific expressions such as GOOD/BAD. I.e. signficantly increased amount in A compared to B instead of better in A compared to B.

Round 2

Reviewer 1 Report (Previous Reviewer 2)

Comments and Suggestions for Authors

Please include islet histology image and data before and after administration of polysaccharides. It is important to assess the therapeutic effects via both pancreas and colon or colon only. Please define organ index in Materials and Methods.

Author Response

Comments 1: Please include islet histology image and data before and after administration of polysaccharides. It is important to assess the therapeutic effects via both pancreas and colon or colon only. Please define organ index in Materials and Methods.

Response 1: Thank you very much for your suggestions. We agree with your viewpoints and have added a definition of organ index in the Materials and Methods section. In addition, immunohistochemistry of pancreatic tissue was added in the Results section, which demonstrated that GLP-1 can transfer from mice colon tissue to pancreatic tissue, providing a possibility for GLP-1 to activate related pathways or signals in pancreatic tissue. And according to relevant references [1], this will improve the expression levels of related pathways in mice pancreatic tissue and promote the survival of mice pancreatic cells. Unfortunately, given that immunohistochemistry must be performed by euthanized mice, we are unable to provide tissue images before gavage, and due to time constraints, we are unable to supplement this data nor futher pancreatic histology research within the deadline. However, the newly added content includes pancreatic tissue images of healthy mice in DC group and T2DM model mice in NC group, and the effect of polysaccharides in mice pancreatic tissue can be seen through comparison. (The changes in the manuscript have been highlighted.)

Thank you again for your valuable comments. Based on our understanding, our team has carefully revised the manuscript and hopes it has reached the level of publication. If there are any discrepancies with your expectations, we sincerely expect your valuable suggestions and will continue to do our best to improve the manuscript.

Reference
1.Christina N. H.; Louise M.; Lee Y.S.; Julia S.; Anna H.G.; Randy J. S.; Daniel J.D.; FredrikB.; Louise E. O. The gut microbiota regulates hypothalamic inflammation and leptin sensitivity in Western diet-fed mice via a GLP-1R-dependent mechanism. Cell Reports, 2021, 35, 8. https://doi.org/10.1016/j.celrep.2021.109163.

This manuscript is a resubmission of an earlier submission. The following is a list of the peer review reports and author responses from that submission.

Round 1

Reviewer 1 Report

Comments and Suggestions for Authors

The paper by Zhang et al examined the anti-diabetic effect of a polysaccharide extract (Physalis alkekengi L.) using a high-fat high-sucrose + STZ model of diabetes (T2DM).  The authors report that the extract, given at three doses, was as effective as metformin to prevent biochemical characteristics of T2DM.  For most measures, the effect of the extracts on preventing biochemical changes induced by HFHS feeding were modest (antioxidant enzymes).  The methods section lacks essential detail:

-there needs to be a description of the diet and % given in the formulation.

- was anything measured to normalize the polysacchardies to adjust the dose, or was it just by weight.

-food intake data needs to be shown

- T2DM is characterized by weight gain - none of the mice gained weight compared to the control

- why gavage; why not add it to the diet.  How do these doses relate to humans?

- how does one record mental state or fur characteristics - what are the standards

- no description of the statistical analysis except the p-value

Poor grammar throughout the paper (e.g., "1 mice died")

Comments on the Quality of English Language

Poor grammar (as stated above) and very choppy.

Author Response

Thank you very much for your comments on the manuscript. I have responded to your comments, please see the attachment.

Reviewer 2 Report

Comments and Suggestions for Authors

This study revealed that administration of Physalis alkekengi L. improve blood glucose via relief of insulin resistance. However, this study shows only biological reaction using animal. No data about the mechanism. Please discuss to include histology, gene and protein examinations. As minor, the grouping is unclear in Figure 2.

Author Response

(The authors gave the same response as above.)

Round 2

Reviewer 1 Report

Comments and Suggestions for Authors

Many of the items and concerns expressed in the original review have not been adequately done. 

-The entire diet should be described - not just sucrose and fat (35% sucrose in teh control diet is quite high - but need to see everything else, like cornstarch, protein, etc). 

-Table 1 does not contain food intake data - in fact, Table 1 and 2 are the same. 

-The foot notes describing multiple comparisons is fine - but the Stats description in the methods is woefully incomplete -  1-way ANOVA?  2-way?  What type of multiple comparison test?  Unequal variance?  etc

Comments on the Quality of English Language

Moderate issues.

Reviewer 2 Report

Comments and Suggestions for Authors

Unfortunately, the authors did not accept my suggestions including adding the histological and PCR data for the pancreas.
